Life histories predict genetic diversity and population structure within three species of octopus targeted by small-scale fisheries in Northwest Mexico

Domínguez-Contreras José F. fradoco@gmail.com 1 2
Munguia-Vega Adrian airdrian@email.arizona.edu 3 4
Ceballos-Vázquez Bertha P. 2
Arellano-Martínez Marcial 2
García-Rodríguez Francisco J. 2
Culver Melanie 3 5
Reyes-Bonilla Hector 1
1 Departamento Académico de Ciencias Marinas y Costeras, Universidad Autónoma de Baja California Sur , La Paz , Baja California Sur , Mexico
2 Instituto Politécnico Nacional, Centro Interdisciplinario de Ciencias Marinas , La Paz , Baja California Sur , Mexico
3 Conservation Genetics Laboratory, School of Natural Resources and Environment, University of Arizona , Tucson , AZ , United States of America
4 Comunidad y Biodiversidad A. C. , Guaymas , Sonora , Mexico
5 U.S. Geological Survey, Arizona Cooperative Fish and Wildlife Research Unit, School of Natural Resources & Environment, University of Arizona , Tucson , AZ , United States of America
Toonen Robert
Electronic publication date: 2018 Feb 15
Publication date: 2018
Volume: 6
Electronic Location ID: e4295
Received 2017 Jun 10; Accepted 2018 Jan 5
Copyright year: 2018
License: This is an open access article, free of all copyright, made available under the Creative Commons Public Domain Dedication. This work may be freely reproduced, distributed, transmitted, modified, built upon, or otherwise used by anyone for any lawful purpose.
License URL: https://creativecommons.org/publicdomain/zero/1.0/

Keywords: Octopus, Fecundity, Planktonic paralarval duration, Paralarval dispersal, Marine connectivity, Gulf of California

Funding: Secretaria de Investigación y Posgrado 20120971 20121594 20130059 20130089 20140781 20140465 20150998 20150117 CONACyT 108230 CONACyT Fronteras de la Ciencia 292/2016 Consejo Nacional de Ciencia y Tecnología doctoral 328943 postdoctoral 291053 David and Lucile Packard Foundation #2013-39400 #2015-62798 This research was partially financed by Secretaria de Investigación y Posgrado projects: 20120971, 20121594, 20130059, 20130089, 20140781, 20140465, 20150998, 20150117, CONACyT 108230 and institutional fund CONACyT Fronteras de la Ciencia 292/2016. José F Domínguez-Contreras benefited from Consejo Nacional de Ciencia y Tecnología doctoral (328943) and postdoctoral (291053 estancias posdoctorales nacionales 2016-1) scholarships. This work was partially supported via the PANGAS Science Coordination by the David and Lucile Packard Foundation grants #2013-39400, #2015-62798. There was no additional external funding received for this study. The funders had no role in study design, data collection and analysis, decision to publish, or preparation of the manuscript.

==============================
The fishery for octopus in Northwest Mexico has increased to over 2,000 tons annually, but to date the specific composition of the catch has been ignored. With at least three main species targeted by artisanal fisheries in the region with distinct life histories, the lack of basic biological information about the distribution, metapopulation size and structure of each species could impede effective fisheries management to avoid overexploitation. We tested if different life histories of three species of octopus could help predict observed patterns of genetic diversity, population dynamics, structure and connectivity and how this information could be relevant to the sustainable management of the fishery. We sequenced two mitochondrial genes and genotyped seven nuclear microsatellite loci to identify the distribution of each species in 20 locations from the Gulf of California and the west coast of the Baja California peninsula. We tested five hypotheses derived from population genetic theory based on differences in the fecundity and dispersal potential for each species. We discovered that Octopus bimaculoides with low fecundity and direct development (without a planktonic phase) had lower average effective population size and genetic diversity, but higher levels of kinship, population structure, and richness of private alleles, than the other two species. These features indicated limited dispersal and high local recruitment. In contrast, O. bimaculatus and O. hubbsorum with higher fecundity and planktonic phase as paralarvae had higher effective population size and genetic diversity, and overall lower kinship and population structure than O. bimaculoides. These observations supported higher levels of gene flow over a larger geographical scale. O. bimaculatus with the longest planktonic paralarval duration and therefore larger dispersal potential had differences in the calculated parameters possibly associated with increased connectivity. We propose O. bimaculoides is more susceptible to over exploitation of small, isolated populations and could have longer recovery times than the other two species. This species may benefit from distinct fishery management within each local population. O. bimaculatus and O. hubbsorum may benefit from fishery management that takes into account metapopulation structure over larger geographic scales and the directionality and magnitude of larval dispersal driven by ocean currents and population connectivity among individuals of each locality. The distribution of each species and variations in their reproductive phenology is also important to consider when establishing marine reserves or seasonal fishing closures.

Introduction

As fish catches are collapsing around the world, the focus of commercial fisheries has shifted to resources within lower trophic levels, but with similar or higher economic revenues (Pauly et al., 2002; Sala et al., 2004; Watson & Pauly, 2001). Cephalopods are a marine resource with lower trophic levels capable of supporting the substantial expansion of fisheries landings (Arkhipkin et al., 2015; Doubleday et al., 2016). Therefore fishing pressure is expected to increase in the near future as a response to growing demands of marine resources associated with increase of global human population (Hunsicker et al., 2010). Cephalopods account for about 20% of the fisheries landing in the world, mainly comprised of squids (FAO, 2015). Octopus catches targeted by small-scale fisheries have increased considerably from 1970 (∼3,000 ton/year) to 2017 (∼60,000 ton/year) and its commercial value is sometimes higher than squids (FAO, 2015). During 2003–2013 most of the global production (80%) was caught in Mexico (36%), Spain (17%), Portugal (15%), Italy (12%) (FAO, 2015). In Mexico O. maya (Voss & Solís-Ramírez, 1966) is the most economically important species captured along the Atlantic coast (NOM-008-PESC-1993; Jurado-Molina, 2010).

At least 10 different octopus species have been described in the Mexican Pacific, including O. bimaculatus (Verrill, 1883), O. chierchiae (Jatta, 1889), O. digueti (Perrier & Rochebrune, 1894), O. bimaculoides (Pickford & MacConnaughey, 1949), and Berry’s (1953) octopuses: O. alecto, O. fitchi, O. hubbsorum, O. veligero, O. rubescens and O. penicillifer (Brusca, 1980; Gotshall, 1998; Hochberg & Fields, 1980; Norman & Hochberg, 2005; Roper, Sweeney & Hochberg, 1995). Recent studies indicate that likely three species contribute most of the catch volume in the Pacific coast of Mexico, namely O. hubbsorum (Alejo-Plata et al., 2009; Domínguez-Contreras et al., 2013; López-Uriarte, Ríos-Jara & Pérez-Peña, 2005), O. bimaculatus (López-Rocha et al., 2012; Villegas et al., 2014), and O. bimaculoides (González-Meléndez, 2012). In Northwest Mexico, the octopus fishery represents an important income for small-scale fishers that sell the catch locally or in commercial markets (Finkbeiner, 2015; Finkbeiner & Basurto, 2015; Moreno-Báez et al., 2012). However, it still unclear which species contribute to the catch in different localities because official fishery statistics do not attempt to distinguish different species. During 2014, official reports indicate that artisanal fishery operating along Norwest Mexico produced at least ∼2,000 tons of octopus worth ∼350,000 USD (CONAPESCA, 2014). Most capture of octopus in this region takes place in the Gulf of California year-round via hooka diving with an air compressor or using traps. In the past, it was suggested that the fishery might be targeting at least two different species (O. bimaculatus and O. hubbsorum) (Moreno-Báez et al., 2012). The lack of species identification in octopus fisheries reports is due to their dynamic biological behavior and ability to change color, pattern, texture and shape (Boyle & Von Boletzky, 1996). In addition, their anatomy includes few hard structures that makes the species identification to the species level difficult for fishermen and fishery managers, especially in octopods (Hanlon, 1988).

Ignoring which species are being fished, and their geographic distribution, could have serious detrimental consequences in the long-term, not only for local fisheries management but for the conservation of the species (Garcia-Vazquez et al., 2012). For example, without precise fishery monitoring, it could be impossible to estimate if a particular species or stock is being over or under exploited in a certain area (Marko et al., 2004). The problem of not identifying each species could be particularly serious if octopuses show contrasting life histories and population dynamics that may translate into distinct levels of maximum sustainable yield (MSY) and recovery times. Different life histories may require distinct management strategies during different seasons and over differing geographic scales. O. bimaculatus could potentially be sympatric with O. bimaculoides in the west coast of the Baja California peninsula, while O. bimaculatus could potentially overlap its geographic distribution with O. hubbsorum in the Gulf of California (Table 1). The reproductive season is different for each species, and the three species differ in their fecundity, egg size and planktonic paralarval duration (PPD) (Table 1). Octopus bimaculoides spawns hundreds of large eggs and lacks a paralarval stages and therefore does not pass through a planktonic phase. O. hubbsorum lays thousands of smaller-sized eggs and the PPD is probably similar to O. vulgaris based on the size of its eggs (∼60 days, Iglesias et al., 2007). O. bimaculatus lays thousands of medium-sized eggs and shows a relatively longer PPD (up to 90 days) than O. hubbsorum and O. bimaculoides (Table 1). The three species have life spans lasting between 1.5 and two years and males typically have smaller size at sexual maturity than females (Table 1).

Table 1 Life history.

Life history of three species of octopus from Northwest Mexico.

Life history	O. bimaculoides	O. hubbsorum	O. bimaculatus	References	
Geographic distribution	
From CA, USA to Bahia
San Quintin in BC,
Mexico.	
From Bahia Magdalena,
BCS to Oaxaca, including
the Gulf of California.	
From CA, USA to Bahia
Vizcaino BCS, including the
Gulf of California	(2, 3, 4 and 10)	
Reproductive period	Santa Barbara, CA, USA (Dec–May)
San Quintin, BCP, Mexico (Oct–Jan)	Pacific coast of BCP
(May–Oct)
Gulf of California
(Mar, Sep–Dec)	Pacific coast of BCP
(Jan–Jun)
Gulf of California
(Jun–Sep)	(1, 2, 3, 5, and 8)	
aFecundity	Eggs laid in festoons
137–780	Clutch eggs
105,000–144,000
Ripe ovarian eggs
240, 050
(range 22,447–545,444)	Clutch eggs
>20,000
Ripe ovarian eggs
91,407 ± 75,361 SD (range 11,618–372,269)	(1, 2, 6, 9, 11 and 12)	
aEgg size (length) and ripe ovarian eggs size	10–12 mm
(range 9.5–16 mm)	1.66 ± 0.74 mm
Ripe ovarian eggs 2.07 mm (range 0.7–3.7 mm)	4–7 mm
Ripe ovarian eggs
(range 1.8–4 mm)	(1, 2, 3, 9, 11, and 12)	
Planktonic larval duration (paralarvae)	Absent, direct development to juvenile, benthic hatchlings	Present but the time is uncertain (Probably ∼60 days)	2–3 months (60 to 90 days)	(1, 2, 3, and 11)	
Size at sexual maturity	55 mm (ML) males
110 mm (ML) females	70 mm (ML) males
119.7 mm (ML) females	124.5 mm (ML) males
147.0 mm (ML) females	(2, 6, 8, and 12)	
Lifespan (years)	Short (1.0–1.5)	Short (1.5)	Short (1.5–2.0)	(2, 3, and 6)	
Notes.

a considering average, min and max reported value. (1) Ambrose (1981), (2) Forsythe & Hanlon (1988a), Forsythe & Hanlon (1988b) (3) Ambrose (1990), (4) López-Uriarte, Ríos-Jara & Pérez-Peña (2005), (5) Castellanos-Martínez (2008), (6) López-Uriarte & Rios-Jara (2009), (8) Domínguez-Contreras (2011), (9) Cardenas-Robles (2013), (10) Domínguez-Contreras et al. (2013), (11) Alejo-Plata & Herrero-Alejo (2014) and (12) Alejo-Plata & Gómez-Márquez (2015).

BCP Baja California Peninsula

ML Mantle Length

Our main hypothesis is that differences in the life history among the three octopus species from Northwest Mexico could translate into distinct patterns of genetic diversity, population dynamics, structure and larval connectivity relevant for sustainable fisheries management. We used two mitochondrial DNA markers and seven nuclear microsatellite loci, informative for the three species, to infer relevant differences in population parameters and evolutionary processes among species. We first established the geographic distribution of each species through genetic identification of tissue samples collected over the study area. Later, five a priori hypotheses were tested based on the fecundity and potential for paralarval dispersal reported in the scientific literature for each species derived from theoretical and empirical population genetic studies regarding expected effective population size, genetic diversity, genetic relatedness within populations (kinship) and population structure (Tables 1 and 2). We discuss implications of our results for fisheries management of the three species in Northwest Mexico.

Table 2 Hypotheses.

Hypotheses regarding levels of genetic diversity and structure based on the life history of three species of octopus from Northwest Mexico.

Hypotheses	O. bimaculoides	O. hubbsorum	O. bimaculatus	References	
Effective population size (Ne)	Small	Medium	Large	(1 and 2)	
Genetic diversity (allelic richness) (NE, RA)	Low	Medium	High	(1 and 2)	
Diversity of private alleles (RPA)	High	Medium	Low	(3 and 4)	
Genetic structure (FST)	High	Medium	Low	(5, 6, and 7)	
Genetic relatedness (R)	High	Medium	Low	(8 and 9)	
Notes.

(1) Romiguier et al. (2014), (2) Ellegren & Galtier (2016), (3) Beger et al. (2014), (4) Munguía-Vega et al. (2015), (5) Selkoe & Toonen (2011), (6) Riginos & Liggins (2013), (7) Selkoe et al. (2014), (8) Christie et al. (2010), (9) Burgess et al. (2014).

Materials & Methods

Sample collection and DNA extraction

We obtained 316 tissue samples of octopus (arm tissue) collected between 2008 and 2013 from 20 localities in both coasts of the Baja California peninsula, including the Northeast coast of the Gulf of California (Field experiments were approved by Secretaría de Agricultura, Ganadería, Desarrollo Rural, Pesca y Alimentación, SAGARPA No. PPF/DGOPA.09151.260809.2885 and PPF/DGOPA-224/16) (Fig. 1). The field sampling took place mainly during spring and summer (Table S1) at fishing communities with the help of small-scale fishers. Octopuses were collected at seven locations along the west coast of the Baja California peninsula, (Ejido Erendira close to Ensenada, Baja California down to El Conejo in Baja California Sur) and 13 sites from the central (Santa Rosalía) and northern Gulf of California (from the northern tip of Bahía de Los Angeles and Isla Tiburón up to Puerto Peñasco), including the Midriff islands. The Midriff islands include many islands and islets in the northern Gulf of California (Fig. 1). Several locations are remote and with difficult access, therefore had smaller samples sizes, while others localities with low number of samples was due to the difficulty of catching octopuses outside their reproductive season. We distinguished between O. bimaculatus and O. bimaculoides based on distinctive characteristics of the gonads of mature females using criteria described by Pickford & MacConnaughey (1949). O. hubbsorum was identified using morphological traits described by Domínguez-Contreras et al. (2013); and original descriptions of Berry (1953). Tissue samples were stored in 96% ethanol in the field and −20 °C in the lab. We extracted DNA using the DNeasy blood and tissue kit (QIAGEN, Valencia, CA, US) following the manufacturer specifications.

Figure 1 Study area.

Locations of 20 octopus populations sampled from Northwest Mexico. B.C, Baja California; B. C. S, Baja California Sur; NGC, Northern Gulf of California. The blue stars represent main fishing localities, and the red circle represents the Midriff Island region.

Mitochondrial DNA sequencing

We amplified two fragments of mitochondrial genes for a subset of the samples to detect the presence of each species at representative locations and to control for the differential success of cross-amplifying microsatellite loci (see below). We selected 97 individuals from 13 localities, including 8 individuals per locality except for Puerto Refugio which had one sample analyzed. We targeted the large ribosomal subunit rRNA (16S) gene employing primers L1987 5′-GCCTCGCCTGTTTACCAAAAAC-3′ and H2609 5′-CGGTCTGAACTCAGATCACGT-3′ (Palumbi et al., 1991) and the Cytochrome Oxidase subunit 1 (COI) gene with primers LCO 1490 5′-GGTCAAACAAATCATAAAGATATTGG-3′and HCO2198 5′-TAAAATTCAGGGTGACCAAAAAATCA-3′(Folmer et al., 1994). We used 25 µL volume PCRs reactions with 15–40 ng genomic DNA, 1 × PCR buffer, 0.2 mM each dNTP, 2 mM MgCl2, 0.2% BSA, 1 U Taq DNA polymerase (Invitrogen, Carlsbad, CA, USA) and 0.5 µM of each primer for both markers. PCR thermo-cycling consisted of denaturation at 94 °C for 2 min, 30 cycles of 94 °C for 1 min, annealing at 51 °C (COI) or 45.5 °C (16s rDNA) for 1 min, and extension at 72 °C for 2 min, followed by a final extension of 72 °C for 7 min. PCR products were purified using ExoSAP (Affimetrix, INC; Santa Clara, CA, USA) and both strands were sequenced on an Applied Biosystems 3730XL DNA Analyzer (Applied Biosystems, Foster City, CA, USA) at the University of Arizona Genetics Core (UAGC).

Genotyping of microsatellites markers

We tested 15 microsatellite markers developed for O. bimaculatus (Domínguez-Contreras et al., 2014) and based on PCR amplification success selected seven unlinked microsatellites (Ocbi25, Ocbi35, Ocbi39, Ocbi41, Ocbi47, Ocbi48, and Ocbi50) that were polymorphic and informative among the three octopus species. We genotyped 316 samples following PCR methods previously described (Domínguez-Contreras et al., 2014). PCR products were sized on an Applied Biosystems 3730XL DNA Analyzer at the University of Arizona’s UAGC Core Facility. Microsatellite electropherograms were scored using GeneMarker Version 2.6.0 (SoftGenetics LLC, State College, PA, USA). Allele sizes were assigned into bins using FLEXIBIN (Amos et al., 2007). Deviations from Hardy-Weinberg equilibrium (HWE) were estimated using GENEPOP 4.2 (Raymond & Rousset, 1995). We used MICROCHECKER 2.2.3 to test for genotyping errors and presence of null alleles (Van Oosterhout et al., 2004).

Species identification

We used the mitochondrial sequences and microsatellite genotypes to identify species for each sampled individual using phylogenetic analyses of sequence data and Bayesian assignment analyses of microsatellite genotypes. The 16S rDNA and COI sequences were edited using Chromas Pro Version 1.6 and aligned using MUSCLE multiple alignment tools implemented in Mega6 (Tamura et al., 2013). We used JmodelTest 2 (Darriba et al., 2012; Guindon & Gascuel, 2003) to select the best fit model of nucleotide substitution for phylogenetic analyses using the Akaike and Bayesian information criteria. We applied the Jukes-Cantor (JC) model with 1,000 bootstraps to estimate genetic distances and constructed a Neighbor-joining (NJ) tree using 10,000 bootstraps replications in MEGA (Tamura et al., 2013).

STRUCTURE version 2.3.4 (Pritchard, Stephens & Donnelly, 2000) was used to analyze microsatellite genotypes using admixture and without prior location information, with allele frequencies correlated among populations. We used a duration of the burnin period of 1 × 106, a number of MCMC repeats after burnin of 2 × 106, with 10 iterations for each number of genetic clusters (K), and K assumed to vary between 1 and 20. To determine the optimal number of K, we selected the number of clusters by looking at the highest likelihood values (mean of 10 iterations) as well as the highest ΔK value implemented in the online software CLUMPAK (Kopelman et al., 2015). We used both values because some evidence has suggested the likelihood method is not always accurate (Evanno, Regnaut & Goudet, 2005). The value of ΔK is based on the rate of change in the log probability of data between successive K values, which provides a relatively better estimate of the number of genetic clusters (Evanno, Regnaut & Goudet, 2005). We used the following criteria to assign individuals to species according the their microsatellite genotypes: First, we excluded those samples that showed missing data at two or more loci. Second, we used a majority rule requiring at least 2/3 (66.66%) of the probability of assignment to any of the three species, and excluded those individuals where this criterion was not met. Third, we only included individuals where the microsatellite and the mitochondrial data agreed on species assignment.

Genetic diversity and effective population size within species

The neutral theory of molecular evolution predicts that in a panmictic population of constant size genetic diversity should be proportional to the effective size of the population (Kimura, 1983). This is because in an idealized, panmictic, population the rate of loss of neutral alleles via genetic drift is inversely proportional to the population size (Charlesworth, 2009). Based on recent comparative studies of octopuses, we expect that species with high brood sizes to produce relatively small eggs (O. bimaculatus and O. hubbsorum) and will have higher genetic diversity and effective population size than species with comparatively low-fecundity that produce a small number of relatively large eggs (O. bimaculoides) (Table 2) (Ellegren & Galtier, 2016; Romiguier et al., 2014). We calculated the number of alleles (NA), effective number of alleles (NE, which takes into account different sample sizes among localities), expected heterozygosity (HE) and observed heterozygosity (HO) with GENALEX 6.501 (Peakall & Smouse, 2012) to evaluate genetic diversity from the microsatellite data. Allelic richness (RA) was estimated using HP-Rare to correct for differences in sample size among locations (Kalinowski, 2005).

Private alleles (alleles that are unique to one population) are expected to be more frequent in genetically isolated populations, while their frequency should be considerably lower in well-connected populations (Beger et al., 2014; Munguía-Vega et al., 2015). If we extend this process to populations within each species, then populations of species with limited opportunities for dispersal (direct ontogenetic development, O. bimaculoides) should show higher frequency of private alleles than species with a pelagic paralarval stage (Table 2). Private allelic richness (RPA) was estimated using HP-Rare to correct for different sample sizes. We estimated a global contemporary effective size (Ne) for each species via the linkage disequilibrium method with a bias correction and a lower allele frequency of 0.05 and 0.02, and with the molecular coancestry method as implemented in the software NE-ESTIMATOR V2 (Do et al., 2014).

Genetic structure within species

Species with a long PPD are expected to disperse in a larger area than species with brief or absent PPD (e.g., direct ontogenetic development) (Shanks, 2009). Consequently, O. bimaculoides with direct development (PPD = 0) should show higher genetic structure (e.g., global FST) (Riginos & Liggins, 2013), than O. hubbsorum with short PPD and O. bimaculatus with long PPD (Table 2) (Selkoe et al., 2014; Selkoe & Toonen, 2011). We conducted a hierarchical analysis of molecular of variance (AMOVA) using 999 permutations in GENALEX 6.501 (Peakall & Smouse, 2012) to estimate the genetic differences observed within and among populations; in other words to estimate genetic structure. We used FreeNA to measure the effect of null alleles on FST estimates of population structure, taking into account the frequency of null alleles estimated with the expectation maximization method (EM) (Chapuis & Estoup, 2007).

Genetic relatedness within populations of each species

The magnitude of local paralarval retention, or the proportion of paralarvae produced within a site that remain in that site, is expected to increase the degree of genetic relatedness (R) within populations (Burgess et al., 2014; Christie et al., 2010). We expect that species with direct ontogenetic development (PPD = 0, O. bimaculoides) should have a higher probability for individuals to remain near their hatching site, and thus to show higher levels of genetic relatedness or kinship within populations than the other two species with a planktonic paralarval drift (Table 2). Since local retention is expected to decrease with increasing PPD (Byers & Pringle, 2006), we expect that genetic relatedness within populations will be lower in the species with the longest known PPD (O. bimaculatus). We used Queller & Goodnight (1989) relatedness metric to calculate pairwise relatedness to describe the number of alleles shared between pairs of individuals and then calculated the average within each population as implemented in GenAlex 6.2 (Peakall & Smouse, 2012). Statistical significance was assessed by 9,999 permutations and 10,000 bootstraps to estimate 95% confidence intervals around the hypothesis of random mating.

Results

Species identification

A total of 1,054 bp were sequenced for 97 individual samples, including 473 bp from the 16S rRNA gene and 581 bp from the COI gene (GenBank Accession numbers KY985098 –KY985194 for 16S, and KY985005 –KY985097 for COI). The optimum model of substitution according to the Akaike and Bayesian criteria was JC for both 16S rRNA and COI. The resulting NJ of 16S rRNA and COI genes showed the monophyletic status of the three species O. bimaculatus, O. bimaculoides and O. hubbsorum (Fig. 2A). O. bimaculoides was present in locations from the west coast of Baja California Peninsula (Ejido Erendira, San Quintin, and Bahía Magdalena), but absent in the Gulf of California. O. bimaculatus was present at only one locality from the west coast of the Baja California Peninsula (Malarrimo) and in samples from the northern Gulf of California including Puerto Peñasco, Puerto Refugio, Puerto Lobos, San Luis Gonzaga, Bahía de los Ángeles and only one individual from Puerto Libertad evidenced with 16S rRNA, (no data was obtained for the COI sequence of this individual). O. hubbsorum was present in several localities from the northern Gulf of California (Puerto Libertad, Isla San Lorenzo, and Bahía Kino) and in the Central Gulf of California (Santa Rosalía) (Fig. 2A). Nucleotide divergence between the three species ranged from 3.3–7.1% for the 16S rRNA gene and from 6.3–10.4% for the COI gene (Table 3). Octopus bimaculoides showed less divergence with O. bimaculatus (3.3% and 6.3%, respectively) than with O. hubbsorum (6.3% and 10.0%, respectively). The largest divergence was observed between O. bimaculatus and O. hubbsorum (7.1% and 10.4%, respectively).

Figure 2 Genetic assignment of octopus samples from fishing localities in Northwest Mexico to three species.

Locations used for both 16s rDNA and COI are indicated with stars (ê). All locations were used for microsatellites analysis. (A) Neighbor-joining trees constructed with 97 haplotypes for both 16s rDNA and COI for O. bimaculatus (blue), O. bimaculoides (purple) and O. hubbsorum (orange). Bootstrap support >99% in 1,000 replicates are shown for branches separating the three species. (B) Bayesian clustering analysis from STRUCTURE showing the probability of individual membership to three genetic clusters (K = 3, 316 individuals). (C) Distribution of octopus species in 20 localities from Northwest Mexico according to phylogenetic and clustering analyses.

Table 3 Nucleotide divergence of both: 16s rDNA gene and COI gene.

Nucleotide divergence between species of octopus identified through the analysis of both the 16s rRNA gene (below the diagonal) and COI gene (above the diagonal). Standard error estimates are shown in parentheses.

	O. bimaculoides	O. bimaculatus	O. hubbsorum	
O. bimaculoides	–	0.0632 (±0.0104)	0.1005 (±0.0142)	
O. bimaculatus	0.0328 (±0.0079)	–	0.1042 (±0.0139)	
O. hubbsorum	0.0629 (±0.0113)	0.0708 (±0.123)	–	

We genotyped seven microsatellite loci in 316 individuals collected from 20 localities and observed an average frequency of missing data of 3.75% (range 1.26–7.27) by locus, and 3.84% (range 0–28.5) by octopus individual. Hardy-Weinberg tests suggested significant deviations at only 7 out of 140 unique loci/location combinations tested without any clear pattern observed within locations or species (after Bonferroni correction P = 0.00036). Only Ocbi39, Ocbi41 and Ocbi50 significantly deviated in 1, 2 and 4 locations from the 20 locations tested, respectively (P = 0.00036). The loci Ocbi41 and Ocbi50 were monomorphic in 1 and 6 localities, respectively (Table S2). Except for the loci Ocbi35 and Ocbi41, the rest of the loci showed null alleles in at least one locality, with Ocbi39 showing null alleles in eight localities. The EM method showed that average frequency of null alleles among all loci/locations varied from 0.000–0.108, for O. bimaculatus 0.025 (range 0.000–0.255), for O. bimaculoides 0.026 (range 0.000–0.265), and for O. hubbsorum 0.041 (range 0.000–0.315) (Table S3).

Before assigning individuals to species in order to test our five a-priori hypotheses we excluded individuals that did not meet our criteria. We excluded 17 samples that showed missing data at two or more microsatellite loci. In our dataset, 92.78% of individuals assigned to one species using 16S rRNA and COI sequences (Fig. 2A) were correctly assigned to the same species using microsatellite genotypes (Fig. 2B). However, we found 20 individuals that did not comply with the 2/3 rule of ancestry to a single species according to the nuclear genome and were excluded from further analyses. These individuals showed a shared ancestry between O. hubbsorum and O. bimaculatus, mainly in the localities of Puerto Peñasco, Puerto Refugio and Puerto Libertad (Table S4). These locations are within the limit of the geographic range between the two species (Table S5). In Puerto Peñasco, two cases were observed in which the mtDNA identified the individuals as O. bimaculatus, whereas their microsatellite ancestry assigned them to O. hubbsorum (Table S5).

The STRUCTURE analyses showed a modal frequency that supported the presence of at least two clusters or species (ΔK = 2, Fig. S1A) according to the ΔK method (Evanno, Regnaut & Goudet, 2005). However, the highest mean value of the ln probability of data for K = 2 (average ln[K] =  − 8362.29, Fig. S1B) was close to K = 3 (average ln[K] =  − 8086.16, Fig. S1B) in 10/10 repetitions, and in both cases the matrix of similarity scores produced by Clumpak among runs aligned were identical 0.999 (Fig. S1C). The STRUCTURE bar plots (Fig. 2B) showed that K = 3 clearly distinguished the three clusters or species previously identified in the phylogenetic analyses of the mitochondrial markers and corresponding to O. bimaculoides (N = 36), O. bimaculatus (N = 140) and O. hubbsorum (N = 101) among the 20 localities from Northwest Mexico (Fig. 2B). Based on the STRUCTURE analyses, O. bimaculoides is found almost exclusively in the west coast of the Baja California peninsula (Ejido Erendira, San Quintin, and Bahía Magdalena), while the analyses suggested a few individuals inside the Gulf of California were assigned to this species.

Octopus bimaculatus and O. hubbsorum were present in all the area of study. Octopus bimaculatus was collected at La Bocana, Las Barrancas and Malarrimo, and O. hubbsorum in El Conejo along the west coast of Baja California peninsula. In the Gulf of California, O. bimaculatus was collected at Puerto Peñasco, San Luis Gonzaga, Isla Smith, Bahía de Los Angeles and Puerto Lobos. Octopus hubbsorum was present in Puerto Libertad, Isla San Lorenzo, Isla Tiburon, Bahía Kino, and Santa Rosalía (Fig. 2C). STRUCTURE analyses suggested the presence of individuals of O. bimaculatus and O. hubbsorum at Las Barrancas in the west coast of Baja California peninsula and Puerto Peñasco, Puerto Refugio and Isla Tiburón in the northern Gulf of California (Figs. 2B and 2C).

Genetic diversity and effective population size within species

The seven loci were polymorphic for the three species (Table 4). Results generally supported our prediction about higher allelic diversity and effective size in highly fecund species with small eggs (O. bimaculatus and O. hubbsorum) than in species that are less fecund and have larger egg sizes (O. bimaculoides). We observed lower average levels of allelic diversity in O. bimaculoides (NE = 3.62  ±  0.47, RA = 4.50 ±  0.48) than in O. hubbsorum (NE = 5.02  ±  0.53, RA = 4.54  ±  0.12), while results for O. bimaculatus were mixed and showed the largest diversity of effective alleles (NE = 5.64  ±  0.28), and the lowest allelic richness (RA = 4.14  ±  0.07).

Table 4 Genetic variation within populations of three species of octopus.

Sample Size (N), Mean ± Standard Error (SE) of the number of alleles (NA), effective alleles (NE), and observed (HO), expected (HE) heterozygosities, alellic richness (RA) and private allelic richness (RPA).

Species	Population	N	NA	NE	HO	HE	RA	RPA	
Octopus	Ejido Erendira	13	5.00 ± 0.93	3.08 ± 0.52	0.78 ± 0.08	0.61 ± 0.07	3.97 ± 0.64	1.06 ± 0.27	
bimaculoides	San Quintín	9	6.14 ± 1.49	4.44 ± 1.18	0.52 ± 0.12	0.62 ± 0.11	5.46 ± 1.23	3.28 ± 1.54	
	Bahía Magdalena	9	4.29 ± 0.71	3.34 ± 0.62	0.91 ± 0.05	0.65 ± 0.05	4.08 ± 0.65	2.33 ± 0.63	
	Mean ± SE		5.00 ± 0.61	3.62 ± 0.47	0.74 ± 0.06	0.63 ± 0.04	4.50 ± 0.48	1.60 ± 0.48	
Octopus	Puerto Libertad	9	5.00 ± 1.31	4.09 ± 1.07	0.65 ± 0.12	0.59 ± 0.12	4.28 ± 0.98	0.53 ± 0.31	
hubbsorum	Isla San Lorenzo	19	5.71 ± 1.11	3.87 ± 0.83	0.62 ± 0.14	0.61 ± 0.10	4.43 ± 0.79	1.59 ± 0.56	
	Isla Tiburón	24	7.57 ± 2.07	5.15 ± 1.53	0.58 ± 0.15	0.61 ± 0.13	4.46 ± 0.96	0.51 ± 0.24	
	Bahía Kino	31	8.86 ± 2.22	5.88 ± 1.50	0.52 ± 0.12	0.67 ± 0.10	4.91 ± 0.91	0.49 ± 0.18	
	Santa Rosalía	8	9.86 ± 2.84	6.31 ± 1.79	0.70 ± 0.14	0.66 ± 0.13	4.78 ± 1.00	0.33 ± 0.24	
	El Conejo	8	6.57 ± 1.51	4.82 ± 1.14	0.75 ± 0.12	0.66 ± 0.11	5.00 ± 0.99	0.69 ± 0.21	
	Mean ± SE		7.26 ± 0.79	5.02 ± 0.53	0.64 ± 0.05	0.64 ± 0.04	4.54 ± 0.12	0.69 ± 0.19	
Octopus	La Bocana	4	5.29 ± 0.47	4.36 ± 0.54	0.93 ± 0.07	0.75 ± 0.03	4.35 ± 0.34	0.15 ± 0.04	
bimaculatus	Las Barrancas	3	3.71 ± 0.52	3.23 ± 0.45	0.81 ± 0.14	0.61 ± 0.10	3.71 ± 0.52	0.14 ± 0.06	
	Malarrimo	31	11.43 ± 0.81	6.05 ± 0.82	0.80 ± 0.08	0.79 ± 0.06	4.08 ± 0.30	0.40 ± 0.15	
	Puerto Peñasco	21	10.29 ± 1.02	6.67 ± 1.04	0.87 ± 0.07	0.79 ± 0.07	4.21 ± 0.36	0.33 ± 0.08	
	San Luis Gonzaga	8	6.71 ± 1.02	5.20 ± 0.76	0.79 ± 0.14	0.71 ± 0.12	4.02 ± 0.51	0.12 ± 0.06	
	Puerto Refugio	12	8.00 ± 1.02	5.65 ± 0.81	0.68 ± 0.11	0.76 ± 0.08	4.07 ± 0.38	0.33 ± 0.11	
	Isla Smith	25	11.14 ± 1.24	6.76 ± 0.89	0.84 ± 0.06	0.81 ± 0.06	4.25 ± 0.32	0.40 ± 0.09	
	B.de Los Ángeles	14	9.57 ± 0.75	6.20 ± 0.89	0.68 ± 0.10	0.78 ± 0.07	4.23 ± 0.35	0.24 ± 0.04	
	Puerto Lobos	20	10.14 ± 0.86	6.64 ± 0.79	0.80 ± 0.08	0.82 ± 0.04	4.34 ± 0.23	0.44 ± 0.17	
	Mean ± SE		8.48 ± 0.43	5.64 ± 0.29	0.80 ± 0.03	0.76 ± 0.02	4.14 ± 0.07	0.28 ± 0.04	

We observed that the species with direct ontogenetic development (O. bimaculoides) had the largest average frequency of private alleles (RPA = 1.60 ± 0.48), compared to the species with a planktonic paralarval phase (Table 4). The lowest values were observed in O. bimaculatus (RPA = 0.28 ± 0.04), while O. hubbsorum showed intermediate values (RPA = 0.69  ±  0.19).

We observed the largest contemporary effective population size Ne in O. bimaculatus using both the linkage disequilibrium and the molecular coancestry methods (average LDNE = 190–252, MC = ∞), followed by O. hubbsorum (LDNE = 104–131, MC = 28.1. O. bimaculoides had the lowest effective size according to the two methods (LDNE = 10–18, MC = 10) (Table 5).

Table 5 Contemporary effective population size.

Average and 95% confidence intervals for the contemporary effective population size (Ne) for three species of octopus. Locations were pooled according to the results of the genetic assignment of species (Fig. 2). Ne was estimated with two methods, including linkage disequilibrium (LD; lowest allele frequency used 0.05 and 0.02 respectively) and Molecular coancestry (MC).

	LDNE 0.05	LDNE 0.02	MC	
O. bimaculoides	9.7 (6.4–13.9)	17.9 (13.6–24.0)	9.9 (3.6–19.2)	
O. bimaculatus	190.2 (129.2–324.2)	252.7 (182.8–388.8)	∞ (∞–∞)	
O. hubbsorum	104.8 (69.9–181.3)	131.4 (95.7–197.1)	28.1 (6.8–64.3)	

Genetic structure within species

Pooling sampling locations according to species molecular identification (Fig. 1), we found that the microsatellite data AMOVA test supported the prediction that O. bimaculoides with direct ontogenetic development had higher levels of genetic structure (FST = 0.19, P = 0.000), compared to species with pelagic paralarvae (Table 6). Also, we accepted the hypothesis that O. bimaculatus, with the longest PPD, had overall lower genetic structure (FST = 0.09, P = 0.000) compared with O. hubbsorum, with relatively shorter PPD (FST = 0.16, P = 0.000).

Table 6 Analysis of molecular variance (AMOVA) from microsatellite data within three species of octopus from Northwest México.

Species	Source of variation	Variance	df	Sum of squares	Means of squares	Estimated variance	P value	
Octopus	Among Populations (FST)	19%	2	28.808	14.404	0.610	0.000	
extbfbimaculoides	Among Indiv (FIS)	0%	28	56.338	2.012	0.000	1.000	
	Within Indiv (FIT)	81%	31	80.000	2.581	2.581	0.005	
	Total	100%	61	165.145		3.190		
Octopus	Among Populations (FST)	16%	5	87.004	17.401	0.471	0.000	
hubbsorum	Among Indiv (FIS)	11%	93	256.834	2.762	0.308	0.000	
	Within Indiv (FIT)	73%	99	212.500	2.146	2.146	0.000	
	Total	100%	197	556.338		2.925		
Octopus	Among Populations (FST)	9%	8	92.321	11.540	0.293	0.000	
bimaculatus	Among Indiv (FIS)	3%	129	380.349	2.948	0.094	0.003	
	Within Indiv (FIT)	88%	138	381.000	2.761	2.761	0.000	
	Total	100%	275	853.670		3.148		

The frequency of null alleles can affect the estimates of genetic differentiation, decreasing the genetic diversity and overestimating the FST values (Chapuis & Estoup, 2007). Genetic differentiation with (Null FST) and without (FST) null alleles estimated with FreeNA were similar within each species: O. bimaculoides (Null FST = 0.214 and FST = 0.221), O. bimaculatus (Null FST = 0.092 and FST = 0.088) and O. hubbsorum (Null FST = 0.102 and FST = 0.110) (Table S6).

Genetic relatedness within populations of each species

The average genetic relatedness for three octopus species were significantly greater than expectations based on random mating (all values p = 0.000, Fig. 3). We found that O. bimaculoides with direct ontogenetic development (no paralarval planktonic stage) had the highest average relatedness within populations (R = 0.209), followed by O. hubbsorum with intermediate PPD (R = 0.135), while O. bimaculatus with the longest PPD had the lowest mean level of relatedness (R = 0.020).

Figure 3 Relatedness within three octopus species.

Mean pairwise relatedness (R) values (±95% confidence intervals) within three octopus species, compared with bootstrapped upper (blue) and lower (red) 95% confidence intervals assuming random mating (10,000 bootstraps replicates).

Discussion

We analyzed slowly evolving haploid markers (the mitochondrial genes 16S rRNA and COI) and rapidly-evolving, hypervariable, nuclear markers (seven microsatellite loci) to infer the geographic distribution of three molecularly identified species of octopus among 20 fishing localities from Northwest Mexico and corroborated that differences in the fecundity and potential paralarval planktonic drift (or lack thereof) influence genetic diversity and population structure found within each species.

A minimum of 3% genetic divergence in the COI gene is considered a threshold to distinguish species in metazoans (Hebert et al., 2003). We found a higher divergence among the three species (6%–10%), suggesting they are reproductively isolated biological taxa. We observed a smaller nucleotide divergence between O. bimaculoides and O. bimaculatus probably due to their more recent divergence from a common ancestor (Hebert et al., 2003). The three octopus species studied here are the main targets for small-scale fisheries in Northwest Mexico and our results showed that, although their distribution ranges sometimes overlap, most of the 20 surveyed localities had evidence for the presence of a single species fishery, which occur in different habitats. O. bimaculoides distribute in coastal habitats with low wave energy (enclosed bays and coastal lagoons), although this species also lives at 20 m depth in rocky and forests kelp habitats (Forsythe & Hanlon, 1988a; Sinn, 2008). Along the west coast of the Baja California peninsula exist at least 16 coastal lagoons located between Ensenada (Baja California) and Bahía Magdalena (Baja California Sur) (Lankford, 1977), which probably have been colonized by stepping-stone events among distinct lagoons during rafting behavior (Gillespie et al., 2012). Rafting has been documented for O. bimaculoides and O. bimaculatus on floating objects like macroalgae (Thiel & Gutow, 2005). This paralarval dispersal mechanism could explain progressive colonization events that increase the range distribution into favorable habitats. Our study expanded the previously known range distribution of the three species along the west coast of Baja California peninsula ∼800 km southward for O. bimaculoides, ∼400 km southward for O. bimaculatus and ∼150 km northward for O. hubbsorum. O. bimaculatus was restricted to the northern region of the Gulf of California where its distribution might be influenced by the geographic extent of a cyclonic (anti-clockwise) oceanographic gyre that drift paralarvae during its summer spawning period (Castellanos-Martínez, 2008; Marinone et al., 2008; Munguia-Vega et al., 2014). O. bimaculatus seems to show the pattern of disjunct distribution reported for several temperate species of invertebrates and fishes that are present in the northern part of the west coast of the Baja California peninsula, disappear in the southern region of the Gulf of California and reappear in the northern region of the Gulf of California (Bernadi, Findley & Rocha-Olivares, 2003). The distribution range of O. hubbsorum was conceptually redefined here to include the south part of the Midriff Island region in the Gulf of California (López-Uriarte, Ríos-Jara & Pérez-Peña, 2005; Moreno-Báez et al., 2012). Given the low number of individuals assigned to O. bimaculoides in the Gulf of California, we recommend that further surveys for the species be conducted before the range of the species can be confidently expanded into the Gulf of California.

The three species were sympatric along the west coast of the Baja California peninsula around the Bahia Magdalena region, while in the Gulf of California only O. bimaculatus and O. hubbsorum were sympatric around the Midriff Island region. Both regions have been considered transition zones between temperate and tropical species (Briggs, 1974; Briggs & Bowen, 2012; Brusca, 2010). Given our samples were collected mainly during warm season, it is important to consider the possibility that O. bimaculatus and O. hubbsorum could be sharing the same shelters around the Midriff Islands region in different season of the year, with O. bimaculatus being more frequent during the cold-temperate period (October–March), while O. hubbsorum prefers warm-tropical water conditions (April–September). A pattern of alternate presence of the two species with different thermal preferences could explain why the octopus fishery is carried out through the year in the northern Gulf of California (Moreno-Báez et al., 2012). Thus, at several localities in the northern Gulf of California both species could be the main target of the fishery during different seasons, and at least in some localities where samples in our study were assigned to O. bimaculatus (e.g., Puerto Lobos) there have been recent field (October 2016) observations were only O. hubbsorum individuals were recorded (JF Domínguez-Contreras and A Munguía-Vega, pers. comm., 2016), highlighting the need for seasonal data to complement our current understanding of species captured, particularly in localities near the biogeographic transition zones. Individuals with mixed ancestry that could not been assigned to a single species based on our criteria were found at locations were both O. hubbsorum and O. bimaculatus seem to be present in the northern Gulf of California. These mixed individuals could be the result of either low statistical power of the seven loci employed or hybrids between the two species. Hybridization between octopuses has not yet been documented, however, the copulatory behavior between different octopus species has been recorded, possibly due to low mate availability and a short lifespan (Lutz & Voight, 1994). The likely hybridization in Northwest Mexico should be further studied with a larger sample of nuclear markers.

The life history strategies of each of the three octopus species strongly influenced the genetic diversity and structure among species, showing significant differences in population dynamics and paralarval connectivity. O. bimaculoides without a planktonic phase (direct ontogenetic development) had the smallest effective population size and the lowest genetic diversity (effective alleles) and showed higher levels of relatedness within populations, more structure among populations and a higher proportion of private alleles, compared to the two octopus species with a planktonic paralarval stage. These observations suggest that populations of O. bimaculoide s are comparatively smaller and structured at a local geographic scale, and are likely highly denso-dependent upon local recruitment. In contrast, O. hubbsorum and O. bimaculatus have comparatively higher fecundity and with planktonic paralarval drift that increase their dispersal potential and opportunities for gene flow among populations (Villanueva et al., 2016). These results are consistent with our hypotheses about a larger effective population size that is associated to higher levels of effective alleles and lower levels of genetic relatedness within populations, less genetic structure among populations and lower frequency of private alleles. This implies that O. hubbsorum and O. bimaculatus might depend less on local paralarval retention and more on paralarval dispersal among different populations. However, O. bimaculatus had lower levels of genetic differentiation among populations, lower frequency of private alleles that translated into an overall lower allelic richness and lower genetic relatedness within populations compared to O. hubbsorum. In addition, effective alleles and effective population size in O. hubbsorum were lower compared to O. bimaculatus. Although no studies exist about the PPD of O. hubbsorum, our results are consistent with a relatively shorter PPD and less potential for dispersal compared to O. bimaculatus. This agrees with a recent study suggesting that for species with a planktonic phase, the duration of the planktonic phase increases with mantle length at hatchling (O. hubbsorum = 1.2 mm ML O. bimaculatus = 2.6 mm ML (Alejo-Plata & Herrero-Alejo, 2014; Ambrose, 1981; Villanueva et al., 2016).

An inability to easily identify biological species hampers any effort towards their management and conservation (Bickford et al., 2007). The distinct biogeography and habitat distributions along with contrasting life history traits are expected to have strong direct effects on population parameters. These are key biological features for establishing the spatial scale, location and timing of management actions and rates of sustainable fishing for each species. Therefore, is not advisable to continue with the current management that does not differentiate among the three species. O. bimaculoides with a lower effective population size, and with local populations that are mostly self-sustaining and partially isolated from other nearby populations could be highly susceptible to over exploitation, severe bottlenecks and could show long recovery times if fisheries management erroneously considers all populations as a single stock and ignores the biological and ecological relevance of local population dynamics. O. bimaculoides management taking place at the level of local populations has advantages over single stock management, for instance, it would be possible to assign rigorous catch quotes per individual bay. The species with higher fecundity and dispersal potential (O. bimaculatus and O. hubbsorum) may benefit from implementation of management tools that consider metapopulation dynamics on a larger geographic scale and the presence of larval dispersal among populations, identifying key paralarval sources and paralarval dispersal routes during the PPD, spawning and hatching seasons for each species.

A critical consideration for management of the octopus fishery in the northern Gulf of California is the difference in the spawning seasons between O. hubbsorum (spring and fall) and O. bimaculatus (summer) and its relationship to the current patterns (direction and speed) of paralarval dispersal and its impact on source—sink metapopulation dynamics. Ocean current patterns in the northern Gulf of California are highly directional, or asymmetric, driven by a cyclonic (anti-clockwise) gyre during spring and summer (Marinone, 2012; Marinone et al., 2008) when both O. hubbsorum and O. bimaculatus spawn. However, O. hubbsorum also spawns during autumn and winter (JF Domínguez-Contreras and A Munguía-Vega, pers. comm., 2016) when the northern Gulf of California gyre reverses to an anti-cyclonic (clockwise) direction (Lavin & Marinone, 2003; Marinone, 2012), effectively transforming key larval sources during spring-summer into larval sinks during autumn-winter. When implementing spatial management tools in systems with strong asymmetry in the direction of the currents, including marine reserves, it is advised that reserves are located upstream according to the main flow to protect the sources of larvae that support multiple downstream fishing sites (Beger et al., 2014; Munguia-Vega et al., 2014). These observations imply that selection of the location of marine reserves for octopus in the northern Gulf of California must consider the cyclonic phase of the oceanographic gyre for both octopus species and the influence of the currents during the anti-cyclonic phase for O. hubbsorum. Also, temporal fishing closures based on the spawning period of a single species, like the one recently implemented in the northern Gulf of California based on O. bimaculatus (Opinión Técnica No. RJL/INAPESCA/DGAIPP/1065/2015; DOF, 2016, 01 junio), might be only partially effective for protecting the recruitment of the other species present in the same locations but with a different spawning season (O. hubbsorum, López-Uriarte, Ríos-Jara & Pérez-Peña, 2005; Moreno-Báez et al., 2012). Similarly, minimum sizes of capture established based on size at sexual maturity for O. bimaculatus might overestimate the minimum size required for O. hubbsorum (Table 1). Also beneficial is recognizing that growth and reproductive biology in octopus is augmented by higher temperatures and food availability (Forsythe & Hanlon, 1988b). Our findings highlight that sustainable fisheries management will heavily depend upon establishing management tools that match the geographic and habitat distribution, life history and population dynamics of the biological species targeted by multi-specific fisheries.

Supplemental Information

Table S1 Collection of samples by locality and month of sampling

Months in which the samples from 20 localities were collected in northwest Mexico 2008-2013.

Click here for additional data file.

Table S2 Hardy-Weinberg equilibrium

Tests of Hardy-Weinberg equilibrium in 20 populations of octopus genotyped at 7 loci. Estimation of exact P-values by the Markov chain method. Parameters: 10,000 dememorization, 1000 batches and 10,000 iterations per batch.

Click here for additional data file.

Table S3 Null allele frequency

Average null allele frequency obtained with FREENA using the Expectation Maximization algorithm of Dempster 1977 in 20 populations of octopus genotyped at 7 loci.

Click here for additional data file.

Table S4 Shared ancestry among individuals of the three octopus species

Individuals that could not be assigned under the criterion of at least 2/3 of assignment probability to a single species, and that show show assignment probabilities shared between two species. (*) Indicates the samples that were assigned with mtDNA (16s rDNA and COI) and a higher probability of assignment by microsatellites.

Click here for additional data file.

Table S5 Individuals identified in the limits of their geographic distribution

Probability of assignment for individuals identified in the limit of the geographic distribution of the species to which they belong. The values in bold indicate the highest observed assignment probability to a single species (microsatellites). (*) Possible hybrids identified by a mismatch between mtDNA (16s rDNA and COI) and microsatellite assignment.

Click here for additional data file.

Table S6 FST Estimates of genetic differentiation considering null alleles

Estimates of FST values with and without considering null alleles according to the software FREENA. Confidence intervals (CI) calculated using 10,000 bootstraps.

Click here for additional data file.

Figure S1 Structure clustering algorithm

(A) Delta K is the mean of the absolute values, (B) mean and standard deviation of Ln probability of data for No. of genetic cluster (K) = 1 to 20 and (C) bar plot showing the mean individual assignment probabilities among 10 independent replicates of both K = 2 and K = 3.

Click here for additional data file.

Supplemental Information 1 COI sequences

Click here for additional data file.

Supplemental Information 2 COI protein

Click here for additional data file.

Supplemental Information 3 16S rDNA sequences

Click here for additional data file.

Supplemental Information 4 COI and 16S sequences Ids

Click here for additional data file.

Supplemental Information 5 Raw data

Click here for additional data file.

We thank several fishing cooperatives, civil society organizations and fisherman that helped collecting octopus samples: Sociedades Cooperativas de Producción Pesquera: de La Purísima, de Bahía Magdalena y de Puerto Chale, fisherman from San Quintin and Ejido Erendira, Dra. Ivonne Posada, and partners of the PANGAS project including Centro Intercultural de Estudio de Desiertos y Oceános A.C. (CEDO), Comunidad y Biodiversidad A.C, Pronatura Noroeste A.C. and fishing cooperatives from the Northern Gulf of California. Karla Vargas and Stacy L. Sotak helped us at various stages during microsatellite genotyping at the University of Arizona. Jaime Gómez-Gutíerrez provided valuable feedback on earlier versions of the manuscript.

Additional Information and Declarations

Competing Interests

Author Contributions

Field Study Permissions

DNA Deposition

Data Availability

The authors declare there are no competing interests. Any use of trade, firm, or product names is for descriptive purposes only and does not imply endorsement by the US Government.

José F. Domínguez-Contreras and Adrian Munguia-Vega conceived and designed the experiments, performed the experiments, analyzed the data, contributed reagents/materials/analysis tools, wrote the paper, prepared figures and/or tables, reviewed drafts of the paper.

Bertha P. Ceballos-Vázquez, Marcial Arellano-Martínez, Francisco J. García-Rodríguez, Melanie Culver and Hector Reyes-Bonilla contributed reagents/materials/analysis tools, prepared figures and/or tables, reviewed drafts of the paper.

The following information was supplied relating to field study approvals (i.e., approving body and any reference numbers):

Field experiments were approved by Secretaría de Agricultura, Ganadería, Desarrollo Rural, Pesca y Alimentación (SAGARPA).

The following information was supplied regarding the deposition of DNA sequences:

The COI and 16S rDNA sequences described here are accessible via GenBank accession numbers KY985098 to KY985194.

The following information was supplied regarding data availability:

The raw data is provided in the Supplemental Files.

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
