# Peer review of "Life histories predict genetic diversity and population structure within three species of octopus targeted by small-scale fisheries in Northwest Mexico"

_PeerJ, doi:10.7717/peerj.4295_

## Round 0.1 · original submission · Major Revisions

We now have comments back from two referees who are each enthusiastic about your work. On the whole, they find the study to be interesting, well-written and appropriately analyzed, but also point out a couple of issues that need to be addressed prior to publication. In particular, there is some concern about how robust your pairwise Fst values are given the sample sizes, and whether the conclusions drawn from between-locality samples with single digit sample sizes are sufficiently strong to make management recommendations. Further the referees question exactly how you assigned individuals to species for these analyses. For example, what criterion used was in the Structure analysis (cut-off for probability that an individual was assigned to a particular cluster?) to determine species relationships. What was done with individuals that were not clearly assigned to a particular species - the analyses appear to assume that each locality has a single species present, but from Fig.2A and the text, it seems clear that samples are not single-species, and it is not clear to the reader as to what happened to the odd individuals. If I understand your manuscript correctly, most samples contain predominantly a single species, but it needs to be explained clearly in the text so that the reader is not guessing at the methods or results. This also seems at odds with the presentation in Fig.2C, which gives the reader the visual impression that the species distributions do not overlap. Clarifying this issue is of critical importance to the evaluation of the manuscript, because the impression from the readers will be quite different if the analyses have explicitly considered the presence of other species within localities, or just ignored them and analysed the samples from each locality as a single species. As written the referees and I are unable to determine which is the case.

While both referees recommend minor revisions, each has made suggestions for improvement of the manuscript, and the questions of how exactly the data were analyzed and the strength of the conclusions overstepping the data are not minor in my opinion. Thus while such revisions are likely minor in scope, I am returning a decision of major revisions because I feel that these issues must be addressed appropriately before the manuscript is acceptable for publication.

·

Basic reporting

There are some areas, particularly in the Introduction where the language should be improved to make it more clear for all audiences. Some of the grammar is confusing and ambiguous and a native English speaker should double check the wording. (See my comments on the PDF)
Otherwise, the references, background, figures, data, etc are clear and the paper reads well.
One table (Table 1) could benefit from a delineation between the section on data and the hypotheses.

Experimental design

The experimental design and methods are clearly defined and thorough.

It might be beneficial to have a short section providing the reasoning behind using a subset of samples for the mitochondrial analysis.

The morphological differentiation between species is stated in a slightly challenging way. Since only one individual from each species was identified through morphology, it might not even be necessary to include this; especially if the genetic information on each species already exists to indicate reproductive isolation between the three.

Validity of the findings

You could also discuss the literature that suggests size at sexual maturity is extremely variable depending on food availability and temperature in multiple species (Forsythe + Hanlon, 1988 Marine Biology)

Additional comments

The findings and conclusions are sound and well stated. Implications in octopus fishery management are clearly indicated and discussed. This paper adds to the growing literature on octopus fishery concerns and management recommendations.

Reviewer 2 ·

Basic reporting

Clear, unambiguous, professional English language used throughout. Mostly – some odd phrases and occasional wrong words – suggestions added to the pdf.

Intro & background to show context. - yes

Literature well referenced & relevant. - yes

Structure conforms to PeerJ standards, discipline norm, or improved for clarity. - yes

Figures are relevant, high quality, well labelled & described. - yes

Raw data supplied (see PeerJ policy). - yes

Experimental design

Original primary research within Scope of the journal. - yes

Research question well defined, relevant & meaningful. It is stated how the research fills an identified knowledge gap. - yes, clearly defined in the Abstract, revisited in the Introduction.

Rigorous investigation performed to a high technical & ethical standard. - not sure - see the issues raised in General Comments
As only 316 individuals across 3 species, 20 localities and 5 years are used, suitable rigour could be questioned? But, because of fairly clear distinction of species by molecular typing, and the subsequent use of pooled analysis to assess general species-wide parameters, then the dataset is suitable.


Methods described with sufficient detail & information to replicate. – yes – nice explanations linking hypotheses and predictions to tests employed.Two mitochondrial genes and even nuclear microsatellite loci are suitable markers for the objectives

Validity of the findings

The Conclusions are clearly stated, and on the whole justified. There is some speculation, but this is identified.

However, regarding whether the data are robust, statistically sound and controlled there may be a potentially major issue with the identification to species of individuals and their incorporation to subsequent analyses.

What is done with the individuals in Fig. 2B that are not clearly assigned to one of the three clusters? For example, a number of individuals in the Puerto Penasco locality appear to be more assigned to O. hubbsorum rather than O. bimacularus, but this is not indicated in the species distribution map? This is acknowledged in the text (l.291-294), but the colour coding of the figure does not convey this, and so is slightly misleading. Could be rectified by cross-hatching the sample localities that contain individuals from two species?
This then leads on to the question as to what was the cut-off level in the assignment to decide on which species to assign an individual to? Or was one employed? There is no mention of this in the text, but after this point the reader is presented with a series of tests that are calculated from location-based genetic data which presumes that either all individuals within a location have been assigned to a single species and pooled (which would be wrong as Fig.2B appears to indicate that many samples contain individuals that assign to species different from the one predominant in that locality), or that some individuals have been excluded from the analyses because they did not assign to the predominant species in that locality (which is not mentioned or indicated, and so cannot be presumed).
This is a serious potential flaw in all subsequent analyses, because it is not clear whether some individuals that may be from a different species have been included in the within-species tests, which would introduce substantial random variance in the microsatellite genotype and allele frequencies, and so generate high and random values of statistics such as Fst (which might explain the high values in the pairwise tests – see below?). If the authors can confirm to the Editor that suitable steps were taken to identify the species assignment of each individual before subsequent analyses, and then that individuals were partitioned appropriately so that only single-species datasets were used, then the ms can be adjusted with relatively minor changes (a section in the Methods or Results to explain the ID and partitioning of individuals more thoroughly, and adjustment of Fig.2C to reflect the mixed allocations within localities). If robust ID and removal of individuals was not carried out then the dataset needs re-analysis and re-presentation. I am presuming the former for this review.

Also, you might comment on what the “mixed allocation” individuals represent in Fig.2b?

I’m not convinced by the values given (l.327-334) for pairwise between-locality estimates of Fst based on the 7 microsatellite loci. Many of these tests are based on sample sizes in single digits, and so will suffer from random sampling variation. I think that this section, along with the supporting Tables should be removed, as it is pushing the dataset too far and so detracts from the other better-supported sections.

Annotated reviews are not available for download in order to protect the identity of reviewers who chose to remain anonymous.

---

## Round 0.2 · accepted · Accept

The referee with the most serious concerns about the original submission is satisfied with the revisions, so I am happy to accept your manuscript and move it forward.

Reviewer 2 ·

Basic reporting

.

Experimental design

.

Validity of the findings

.

Additional comments

The authors appear to have taken on board the concerns expressed and have adjusted their analyses and/or interpretation and presentation of data to account for this.